# Lycosin-II Exhibits Antifungal Activity and Inhibits Dual-Species Biofilm by *Candida albicans* and *Staphylococcus aureus*

**DOI:** 10.3390/jof8090901

**Published:** 2022-08-24

**Authors:** Jonggwan Park, Hyeongsun Kim, Hee-Kyoung Kang, Moon-Chang Choi, Yoonkyung Park

**Affiliations:** 1Department of Bioinformatics, Kongju National University, Kongju 32588, Korea; 2Department of Biomedical Science, Chosun University, Gwangju 61452, Korea; 3Research Center for Proteineous Materials, Chosun University, Gwangju 61452, Korea

**Keywords:** antimicrobial peptide, *Candida albicans*, *Staphylococcus aureus*

## Abstract

The increase and dissemination of antimicrobial resistance is a global public health issue. To address this, new antimicrobial agents have been developed. Antimicrobial peptides (AMPs) exhibit a wide range of antimicrobial activities against pathogens, including bacteria and fungi. Lycosin-II, isolated from the venom of the spider *Lycosa singoriensis*, has shown antibacterial activity by disrupting membranes. However, the mode of action of Lycosin-II and its antifungal activity have not been clearly described. Therefore, we confirmed that Lycosin-II showed antifungal activity against *Candida albicans* (*C. albicans*). To investigate the mode of action, membrane-related assays were performed, including an evaluation of *C. albicans* membrane depolarization and membrane integrity after exposure to Lycosin-II. Our results indicated that Lycosin-II damaged the *C. albicans* membrane. Additionally, Lycosin-II induced oxidative stress through the generation of reactive oxygen species (ROS) in *C. albicans*. Moreover, Lycosin-II exhibited an inhibitory effect on dual-species biofilm formation by *C. albicans* and *Staphylococcus aureus* (*S. aureus*)*,* which are the most co-isolated fungi and bacteria. These results revealed that Lycosin-II can be utilized against *C. albicans* and dual-species strain infections.

## 1. Introduction

Increasing antimicrobial resistance is a severe threat to public health, warranting the development of effective drugs to combat infections that are caused by multidrug-resistant pathogens, such as bacteria and fungi [1]. The abuse of antibiotics to combat infections has increased, resulting in the development of various multidrug-resistant pathogens [2]. Fungal infections, such as candidiasis, are also becoming increasingly problematic. Three classes of antifungal agents—azoles, polyenes, and echinocandins—are only used in clinical settings. Therefore, an increase in resistant fungal pathogens raises concerns regarding fungal diseases and public health [3].

*C. albicans* is an opportunistic and prevalent fungal pathogen [4]. *C. albicans* can cause nosocomial bloodstream infections with a mortality rate of 40%, despite the use of antifungal agents [5,6,7]. In particular, *C. albicans* can cause superficial infections in immunocompromised individuals receiving organ transplants and chemotherapy [8]. Moreover, *C. albicans* can form biofilms, which are related to *C. albicans* infections. *C. albicans* biofilms are a community of yeast-form, pseudohyphal, and hyphal cells encased in extracellular polymeric substances (EPS) that are resistant to the host immune system and existing antifungal agents [9]. Resistance to antifungal agents by *C. albicans* biofilms and the ability to colonize surfaces, such as implanted medical devices, negatively impacts patient health [10]. *C. albicans* is the most common fungal pathogen that is regularly found in bacterial-fungal co-infections.

*S. aureus* is a major pathogen that causes clinical infections, such as bacteremia and osteoarticular, skin, pleuropulmonary, and device-associated infections. Moreover, *S. aureus* is frequently isolated with *C. albicans* in bloodstream infections [11]. Importantly, *C. albicans* and *S. aureus* co-infections have increased mortality compared to monomicrobial infections [12]. Therefore, a wide range of novel antimicrobials should be developed to treat these polymicrobial infections.

AMPs play a key role in the innate immune system, functioning as a defense against harmful microbes in various organisms. Additionally, AMPs are a promising alternative to conventional antibiotics. In particular, AMPs yielding broad-spectrum antimicrobial activity and decreased rates of resistance have become potential therapies for the control of infections [13,14].

Lycosin-II, a 21-amino-acid peptide isolated from the venom of the spider *Lycosa singoriensis*, is an α-helix peptide with antibacterial effects against multi-drug resistant nosocomial bacterial pathogens [15]. In a previous study, we confirmed that Lycosin-II, which disrupted the bacterial membrane, was active against oxacillin-resistant *S. aureus* and meropenem-resistant *Pseudomonas aeruginosa* [16]. However, it is not clear how Lycosin-II has antifungal activity and can inhibit the formation of *C. albicans* and *S. aureus* mono- and multispecies biofilms.

In this study, we investigate the antifungal activity and the mechanism of action of Lycosin-II against *C. albicans* by examining membrane depolarization and integrity after exposure to Lycosin-II. Moreover, we confirm that Lycosin-II can inhibit biofilm formation by *C. albicans* and multispecies biofilm formation by *C. albicans* and *S. aureus*.

## 2. Materials and Methods

### 2.1. Materials

Yeast extract dextrose peptone (YPD) broth and Luria–Bertani (LB) broth were purchased from LPS Solution (Daejeon, Korea). Agar and crystal violet were purchased from Duksan (Ansan, Korea). Additionally, Rosewell Park Memorial Institute (RPMI) 1640 medium was purchased from Welgene (Daegu, Korea). Then, calcofluor white, bis(1,3-dibutylbarbituric acid) trimethine oxonol (DiBAC_4_(3)), 2, 7-dichlorofluorescin diacetate (DCFH-DA), SYTOX green, N-acetyl cysteine (NAC), propidium iodide (PI) fluconazole, and amphotericin B were obtained from Sigma-Aldrich (St. Louis, MO, USA).

*C. albicans* (Korea Collection for Type Culture [KCTC] 7270) was used as a reference strain, and it was isolated from a human skin lesion of erosion interdigitalis in Uruguay and purchased from the KCTC. *C. albicans* (Culture Collection of Antibiotics Resistant Microbes [CCARM] 14001, CCARM 14004, and CCARM 14007) isolated in 1999, as well as *C. albicans* (CCARM 14020) isolated in 2002, were used as representative drug-resistant strains and were purchased from the CCARM [17]. *S. aureus* (American Type Culture Collection [ATCC] 25923) was purchased from the ATCC.

### 2.2. Peptide Synthesis, Purification, and Mass Analysis

Lycosin-II and melittin were synthesized, purified, and analyzed as previously described [16]. Briefly, the peptides were synthesized using solid-phase methods with Fmoc (N-(9-fluorenyl)methoxycarbonyl)-protected amino acids in a Liberty microwave peptide synthesizer (CEM Co., Matthews, NC, USA). Coupling was initiated with 0.1 M N-hydroxy benzotriazole (in piperidine), dimethylformamide, 0.45 M 2-(1H-benzotriazole-1-yil)-1,1,3,3-tetramethyluronium hexafluorophosphate (in dimethylformamide), and 2 M N,N-diisopropylethylamine (in N-methyl-pyrrolidone). The peptides were purified by reversed-phase high-performance liquid chromatography with a Jupiter C18 column (250 × 21.2 Zmm, 15 μM, 300 Å; Phenomenex, Torrance, CA, USA). The molecular weight of the peptides was analyzed using a matrix-assisted laser desorption ionization mass spectrometer (Kratos Analytical Inc., Chestnut Ridge, NY, USA).

### 2.3. Antifungal Activity

To determine the antifungal activity of Lycosin-II, melittin, and the antifungal agents, the minimum inhibitory concentrations (MICs) were determined using a broth microdilution method with some modification [18,19]. Lycosin-II, melittin, fluconazole, and amphotericin B were serially diluted two-fold in 10 mM sodium phosphate buffer in 96-well plates. *C. albicans* was cultured overnight at 28 °C in YPD media and then seeded into 96-well plates (2 × 10^4^ colony-forming units (CFUs)/mL; 50 μL/well). After incubation for 16–24 h at 28 °C, the growth of the *C. albicans* strains was determined by measuring the absorbance at 600 nm using a Spectra Max M3 microplate reader. The MICs were defined as the lowest peptide or antifungal agent concentration that could inhibit 90% of the growth of *C. albicans* compared to the negative control.

### 2.4. Time-Kill Kinetics Assay

The time-kill kinetics of Lycosin-II against *C. albicans* (KCTC 7270) were determined. *C. albicans* was grown to mid-log phase in YPD medium at 28 °C. Then, *C. albicans* was diluted in the YPD medium to a final concentration of 2 × 10^4^ CFU/mL. The suspensions were incubated with Lycosin-II at 1× and 2× MICs and with 10 mM sodium phosphate buffer as a control. Next, aliquots were plated on the YPD agar plates and incubated for 16–24 h at 28 °C. After incubation, the CFUs were counted. The survival rate was calculated using the following equation [20]:

Percentage survival rate = (CFUs of *C. albicans* in the Lycosin-II solution–CFUs of *C. albicans* in the control solution) × 100%

### 2.5. Membrane Depolarization Assay

Membrane depolarization in *C. albicans* (KCTC 7270) by Lycosin-II was measured using DiBAC_4_(3) [21]. DiBAC_4_(3) is a membrane-potential-sensitive dye that can enter depolarized cells and increase the fluorescence intensity. Briefly, *C. albicans* was cultured in YPD medium at 28 °C and then washed with phosphate-buffered saline (PBS) three times. The *C. albicans* (5 × 10^6^ CFU/mL) in the PBS was incubated with Lycosin-II at 0.5×, 1×, 2×, and 4× MICs for 3 h at 28 °C. The cells were centrifuged and resuspended in 1 mL PBS supplemented with 2 μg/mL DiBAC_4_(3) for 10 min. The fluorescence intensity of DiBAC_4_(3) was measured using flow cytometry (Beckman, Brea, CA, USA).

### 2.6. PI Uptake Assay

The membrane integrity of *C. albicans* (KCTC 7270) after exposure to Lycosin-II was determined using a PI uptake assay. Propidium iodide is an impermeable dye that is used to determine the permeability of cells by agents [22,23]. *C. albicans* (2 × 10^7^ CFU/mL) in PBS was incubated with Lycosin-II at 0.5×, 1×, 2×, and 4× MICs at 28 °C for 10 min. Then, *C. albicans* was harvested by centrifugation and resuspended in PBS. Next, PI (10 μg/mL) was added to the cells, and these were incubated for 10 min. To remove unbound PI, the cells were washed with PBS. The fluorescence intensity of the PI was measured using flow cytometry (Beckman). Additionally, the PI uptake was visualized using fluorescence microscopy and an EVOS FL Auto 2 imaging system (Invitrogen, Waltham, MA, USA).

### 2.7. SYTOX Green Uptake Assay

To further investigate *C. albicans* membrane damage by Lycosin-II, we used SYTOX green, which is an impermeable dye that interacts with nucleic acids, resulting in enhanced fluorescent intensity. *C. albicans* (KCTC 7270) was washed and resuspended in PBS. Then, *C. albicans* (2 × 10^7^ CFU/mL) in PBS loaded with SYTOX green (1 μM) was treated with Lycosin-II, and the fluorescence was measured using excitation and emission wavelengths of 485 nm and 520 nm, respectively [24]. Additionally, the SYTOX green uptake was visualized using fluorescence microscopy and an EVOS FL Auto 2 imaging system (Invitrogen).

### 2.8. Measurement of ROS

The accumulation of reactive oxygen species (ROS) in *C. albicans* (KCTC 7270) due to Lycosin-II was detected using DCFH-DA [25]. DCFH-DA is oxidized by ROS to dichlorofluorescein (DCF), which emits green fluorescence [26]. *C. albicans* was washed with YPD medium and resuspended at a final concentration of 1 × 10^7^ CFU/mL in YPD medium. Next, suspensions of *C. albicans* were exposed to Lycosin-II at 0.5×, 1×, 2×, and 4× MICs for 1 h at 28 °C. After centrifugation, *C. albicans* was incubated with 10 μM DCFH-DA for 1 h and then washed with PBS. The fluorescence intensity of DCFH-DA was measured using flow cytometry (Beckman).

### 2.9. Analysis of the Effect of NAC on the Antifungal Activity of Lycosin-II

To determine the effect of ROS production on the mode of action of Lycosin-II, we used the ROS scavenger NAC [27]. *C. albicans* (KCTC 7270) was washed and resuspended with PBS at 1 × 10^7^ CFU/mL, and SYTOX green at 1 μM was added to the suspension. Then, NAC was added to the suspension at 5, 7, 7.5, and 10 mM, which was then incubated with Lycosin-II at 1× MIC for 1 h. The fluorescence intensity of the SYTOX green was measured at excitation and emission wavelengths of 485 nm and 520 nm using a Spectramax M3 spectrophotometer (Molecular Devices, Sunnyvale, CA, USA).

### 2.10. Biofilm Inhibition Assays

To analyze the effect of Lycosin-II on *C. albicans* single-species biofilms and polymicrobial biofilms, we used *C. albicans* (CCARM 14020), *S. aureus* (ATCC 25923), and *S. aureus* (ATCC 25923) expressing green fluorescent protein (GFP). We demonstrated in a previous study that the strain CCARM 14020 of *C. albicans* could form a large biofilm [23,28]. *C. albicans* (CCARM 14020) and *S. aureus* (ATCC 25923) were incubated in YPD medium and LB medium. The two strains were washed with RPMI medium supplemented with 2% glucose and adjusted to 1 × 10^6^ CFU/mL for the single-species biofilm evaluation. A total of 180 μL aliquots of *C. albicans* or *S. aureus* and the two mixed strains was added to 96-well plates and incubated with 20 μL Lycosin-II at 0.5–8 μM for 24 h at 37 °C. The supernatant was discarded, and 100% methanol was added to fix the biofilm for 20 min. Then, the methanol was discarded, and the sample was dried completely. The biofilm formation was stained with 0.1% crystal violet for 20 min and then washed three times with distilled water. Next, ethanol (95%) was added to the stained biofilms for dissolution. Biofilm quantification was determined by measuring absorbance at 595 nm using a Versa-Max microplate enzyme-linked immunosorbent assay reader (Molecular Devices) [29,30]. The percentage of biofilm formation was determined using the following equation:Biofilm formation (%) = (A595 of treated biofilm/A595 of untreated biofilm) × 100

### 2.11. Determination of C. albicans and S. aureus Viability in a Dual-Species Biofilm

The biofilms that were formed by *C. albicans* and *S. aureus* were suspended by pipetting. The biofilm suspension was transferred to a 1.5 mL tube, diluted in 1 mL PBS, and vortexed for 5 min [31,32]. Then, YPD and LB agar plates containing vancomycin and amphotericin B were used to distinguish between *C. albicans* and *S. aureus*, respectively. The aliquots were spread on agar plates and incubated at 37 °C for 16 h. The numbers of *C. albicans* and *S. aureus* were determined by counting the CFUs.

### 2.12. Visualization of the Biofilm

To visualize the biofilm after treatment with Lycosin-II, we used calcofluor white (10 μg/mL) to stain *C. albicans* in the biofilm. Moreover, the *S. aureus* (ATCC 25923) strain expressing GFP allowed for the visualization of this bacterium. The images were visualized using EVOS FL Auto 2 fluorescence (Invitrogen).

### 2.13. Statistics

All the data were expressed as three independent experiments using the mean ± standard deviation. Two-tailed Student’s *t*-tests were used to determine the significance of the differences (*, *p* < 0.05; **, *p* < 0.01; ***, *p* < 0.001; ****, *p* < 0.0001; ns, not significant compared to the control).

## 3. Results

### 3.1. Antifungal Activity against C. albicans

The antifungal activities of Lycosin-II, melittin, fluconazole, and amphotericin B against the *C. albicans* strains are summarized in Table 1. Melittin is a lytic peptide with antimicrobial activity that was isolated from Iranian honeybee (Apis mellifera meda) venom and was used as a control in this study. Lycosin-II showed antifungal activity against *C. albicans* with a MIC of 2 μM. Melittin showed antifungal activity against *C. albicans* at concentrations ranging from 2 to 4 μM. We also found that Lycosin-II showed antifungal activity against fluconazole-resistant *C. albicans* strains. A time-kill kinetics assay was performed to determine the time that was required to kill *C. albicans* cells, and Lycosin-II showed antifungal activity in dose- and time-dependent manners. Within 60 min, Lycosin-II at 1× MIC reduced *C. albicans* to >10% viability. In addition, Lycosin-II at 2× MIC was able to completely kill *C. albicans* in 60 min (Figure 1).

### 3.2. Membrane Potential Levels of C. albicans

DiBAC4(3) was used to analyze the effect of the membrane potential in *C. albicans* after exposure to Lycosin-II. *C. albicans* was treated with Lycosin-II at 0.5×, 1×, 2×, and 4× MICs. As shown in Figure 2A,B, 1.58% of the *C. albicans* control group stained positive for DiBAC4(3). However, in *C. albicans* exposed to Lycosin-II at 0.5×, 1×, 2×, and 4× MICs, the fluorescence intensity increased by 5.06%, 24.82%, 90.44%, and 99.04%, respectively, compared to those of the control groups (Figure 2A,B). These results suggested that Lycosin-II has membrane-depolarizing activity against *C. albicans*.

### 3.3. Effect of Lycosin-II on Membrane Integrity

We analyzed whether the integrity of the fungal membrane was damaged by Lycosin-II using PI and SYTOX green. These dyes cannot penetrate intact membranes. However, they interact with nucleic acids after membranes are compromised with antimicrobial agents, thereby increasing the fluorescence intensity. We demonstrated the disruption of *C. albicans* membrane integrity by flow cytometric analysis using PI. Among the control groups, the proportion of *C. albicans* stained with PI was 0.6%, indicating an intact cell membrane. After treatment with Lycosin-II at 0.5×, 1×, 2×, and 4× MICs, the proportions of PI-positive *C. albicans* increased to 2.92%, 30.50%, 89.48%, and 99.92%, respectively (Figure 3A). We further confirmed the integrity of the *C. albicans* membrane using SYTOX green. We also demonstrated that treatment with Lycosin-II induced an increase in the fluorescence intensity of SYTOX in a dose-dependent manner (Figure 3B). Moreover, PI and SYTOX green stained the nucleic acid DNA in *C. albicans* after treatment with Lycosin-II in a dose-dependent manner. Additionally, we confirmed that Lycosin-II damaged the *C. albicans* membranes, causing the uptake of PI and SYTOX green, as demonstrated by the increase in the red and green fluorescence intensities, respectively (Appendix A).

### 3.4. Antifungal Activity against C. albicans

The induction of ROS production by antimicrobial agents is acknowledged as an antifungal mechanism that damages membrane integrity [33]. To analyze intracellular ROS generation by *C. albicans* after Lycosin-II exposure, we used a fluorescent dye, DCFH-DA, which is widely used as an indicator of ROS. We confirmed that Lycosin-II induced ROS production by using flow cytometry to quantify DCF fluorescence. A total of 0.56% of *C. albicans* exhibited ROS-positive staining within the control group. However, dose-dependent ROS generation was observed, with 1.24%, 1.82%, 6.60%, and 27.48% exhibiting ROS-positive staining after treatment with 0.5×, 1×, 2×, and 4× MICs of Lycosin-II, respectively (Figure 4A,B). We used NAC as an ROS scavenger to demonstrate the association between ROS generation due to Lycosin-II and antifungal activity. We confirmed that pretreatment with NAC reduced the fluorescence intensity of SYTOX green (Figure 4C). These results suggested that Lycosin-II induces ROS generation, leading to oxidative and fungal membrane damage.

### 3.5. Inhibitory Effects of Lycosin-II on Biofilm

Biofilms produced by pathogens such as bacteria and fungi worsen disease progression and increase drug resistance [34]. *C. albicans* and *S. aureus* were tested to assess the biofilm inhibitory effect of Lycosin-II. The results show that Lycosin-II at 8 and 2 μM inhibited more than 90% of the biofilm formation by a single species (*C. albicans* or *S. aureus*; Figure 5A,B). Lycosin-II also inhibited the formation of dual-species biofilms containing *C. albicans* and *S. aureus* (Figure 5C). Furthermore, to investigate the viability of *C. albicans* and *S. aureus* in dual-species biofilms, a colony count assay was performed. Lycosin-II resulted in the inhibition of *C. albicans* and *S. aureus* within the biofilm (Figure 5D,E). To visualize biofilm inhibition by Lycosin-II, we used calcofluor white to distinguish the fungi and GFP-expressing *S. aureus*. Treatment with 4 μM Lycosin-II decreased the number of *C. albicans* in the biofilm. Moreover, 2 μM Lycosin-II markedly reduced *S. aureus* in the biofilms (Figure 5F). These results indicate that Lycosin-II effectively inhibited biofilms that contained *C. albicans* and *S. aureus*.

## 4. Discussion

Antibiotics are highly useful for the treatment of infections. However, antimicrobial resistance is increasing globally and is considered a serious threat to public health. The development of antimicrobial agents is a key factor in the treatment of infectious diseases. The increasing incidence of infections by pathogens, such as bacteria and fungi, that are resistant to conventional antimicrobial agents warrants the development of novel antimicrobial agents [35].

Azoles are commonly used to treat Candida spp. infections. Azoles have antifungal activity due to inhibiting the ergosterol biosynthesis pathway and increasing membrane permeability. Recently, many studies have reported that Candida species can be resistant to antifungal agents [36]. Amphotericin B is commonly used to treat fungal infections. However, amphotericin B has side effects, including chills, headaches, kidney damage, and inflammation [37]. Therefore, new antifungal agents should be developed to prevent resistance and toxicity.

Therefore, AMPs offer an effective therapeutic strategy against a wide range of pathogens. The mechanism of action of AMPs is associated with membrane disruption and the targeting of intracellular components, including nucleic acids [38]. Moreover, resistance to AMPs develops slower than it does to conventional antibiotics [39]. Thus, AMPs may be used to overcome antimicrobial resistance.

We previously reported that Lycosin-II, which is derived from the venom of the spider *L. singoriensis*, has antibacterial activity and can act on Gram-negative and Gram-positive bacteria by disrupting the membrane within a concentration range of 1–2 μM. Lycosin-II also exhibited antimicrobial activity against multidrug-resistant (MDR)-*S. aureus* and MDR-*P. aeruginosa*. Moreover, Lycosin-II showed lower cytotoxicity than melittin, which is a known lytic peptide [16]. Lycosin-II also has anti-inflammatory activity by reducing pro-inflammatory cytokines, such as IL-6, IL-8, TNF-α, and IL-1β, in mammalian cells during *S. aureus* and *P. aeruginosa* infections. However, the antifungal activity and mechanism of action of Lycosin-II are not known. Thus, this study aimed to investigate the Lycosin-II antifungal activity, mechanism, and prevention of biofilm formation by *C. albicans* and *S. aureus*.

The MICs were determined to confirm the antifungal activity of Lycosin-II against *C. albicans*. We found that Lycosin-II showed antifungal activity against both standard *C. albicans* (KCTC 7270) and fluconazole-resistant *C. albicans* (CCARM 14001, 14004, 14007, and 14020; Table 1).

The fungal cell wall is an important target in the development of antifungal agents [40]. Therefore, we investigated whether Lycosin-II affects the membrane of *C. albicans*. The mechanisms of action of AMPs include depolarization of the cytoplasmic membrane, which induces membrane permeation [41]. In a previous study, Bac8c, the peptide analog of Ba2A, induced a change in the membrane potential in *C. albicans* [42]. To investigate the mechanism of action of Lycosin-II, we verified if Lycosin-II could depolarize the membrane using DiBAC_4_(3). This dye is a potential sensing probe that can enter a depolarized membrane and bind to intracellular components, increasing the fluorescence intensity. *C. albicans* treated with Lycosin-II showed a greater increase in DiBAC_4_(3) fluorescence intensity than untreated *C. albicans*. These results showed that Lycosin-II induced the depolarization of the plasma membrane of *C. albicans* (Figure 2). Previous studies reported that the membrane of *C. albicans* was damaged by antifungal peptides, such as protonectin isolated from *Agelaia pallipes pallipes* [33]. In contrast, peptides consisting of lysine and tryptophan 4 repeats showed antifungal activity against *C. albicans* but did not affect the membrane integrity [23]. To explore the antifungal mechanism of Lycosin-II in the *C. albicans* membrane, the integrity of the fungal membranes was analyzed using PI and SYTOX green. Both dyes are nucleic acid affinity red and green dyes that enter the fungal cell after membrane integrity is compromised by antimicrobial agents [43]. Therefore, increases in PI and SYTOX green fluorescence intensities indicate a loss in fungal membrane integrity. We confirmed the increased uptake of PI and SYTOX green after Lycosin-II treatment compared to untreated *C. albicans* using flow cytometry and fluorescence spectrophotometry (Figure 3). Moreover, the fluorescence microscopy data showed that Lycosin-II treatment led to increased PI and SYTOX green fluorescence in dose- and time-dependent manners. These data were consistent with the results of the flow cytometry and spectrophotometry (Appendix A). The natural production of ROS plays a role in homeostasis and cellular metabolism. Previous studies have reported the mechanisms of conventional antifungal agents associated with ROS production. The excessive production of ROS induced by antifungal agents affects lipid retention and induces membrane permeability [22,44]. For example, the antifungal agent fluconazole can increase oxidative stress in fungal strains by inducing ROS production [45]. Therefore, we used the ROS probes DCFH-DA and NAC as general scavengers of ROS. Our results showed that Lycosin-II induced ROS generation in *C. albicans* in a dose-dependent manner. In addition, NAC indicated reduced membrane permeability of *C. albicans*, which was caused by Lycosin-II. These results indicated that enhanced ROS production after exposure to Lycosin-II affected membrane permeability in *C. albicans* (Figure 4).

Biofilms are communities of microorganisms that are trapped by EPS and are protected from the host immune system and antibiotics. Microbes in multispecies infections can have altered virulence, proliferation, and antioxidant tolerance. Because microorganisms within mixed fungal–bacterial biofilms have high levels of resistance to antimicrobial agents, fungal–bacterial infections are complex and challenging to treat. *C. albicans* and *S. aureus* are major opportunistic fungi and bacterial pathogens, respectively [46]. Approximately 27% of *C. albicans* infections are polymicrobial with *S. aureus* [47]. *S. aureus* and *C. albicans* exhibit synergistic pathogenicity. For example, *C. albicans* can enhance the resistance of *S. aureus* to vancomycin [48]. Therefore, approaches to combat both bacteria and fungi are urgently needed. In this study, we explored whether Lycosin-II inhibited single-species biofilm formation using a crystal violet assay for biofilm quantification. Additionally, we used calcofluor white and *S. aureus* expressing GFP for biofilm visualization. Lycosin-II prevented biofilm formation by *C. albicans* and *S. aureus* in a dose-dependent manner. Moreover, we confirmed that Lycosin-II inhibited the formation of dual-species biofilms. Additionally, the numbers of live *C. albicans* and *S. aureus* in the biofilms were significantly reduced by Lycosin-II (Figure 5). These results indicated that Lycosin-II had inhibitory activity against dual-species biofilms.

## 5. Conclusions

In conclusion, Lycosin-II had antifungal activity against *C. albicans*. Lycosin-II increased membrane depolarization and permeabilization in *C. albicans*, and Lycosin-II-induced ROS generation was associated with antifungal activity against *C. albicans*. Lycosin-II effectively inhibited the formation of single- and dual-species biofilms. Our results suggested that Lycosin-II may be a useful therapeutic alternative against polymicrobial infections.

## Figures and Tables

**Figure 1 jof-08-00901-f001:**
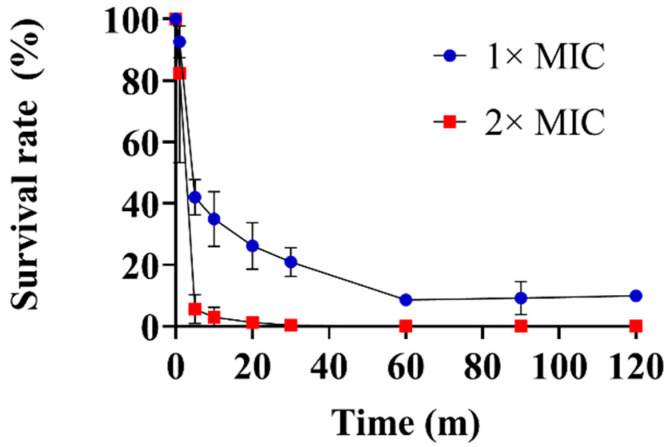
The time-kill kinetics curves of Lycosin-II against *Candida albicans*. *C. albicans* (Korea Collection for Type Culture 7270) was exposed to Lycosin-II for 0, 1, 5, 10, 20, 30, 60, 90, and 120 min. The survival rates were determined by counting the number of colony-forming units at each time point. Abbreviation: MIC, minimum inhibitory concentration.

**Figure 2 jof-08-00901-f002:**
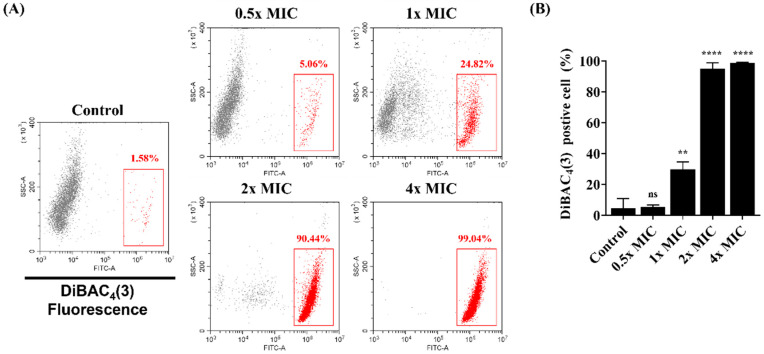
Analysis of *Candida albicans* membrane depolarization after treatment with Lycosin-II. (**A**) Changes in membrane depolarization in *C. albicans* after treatment with Lycosin-II with the peptide at 0.5×, 1×, 2×, and 4× MICs were investigated using flow cytometry with DiBAC4(3). (**B**) The statistical analyses are displayed as averages (*n* = 3). The graph is expressed as the mean ± standard deviation. Significance: **, *p* < 0.01; ****, *p* < 0.0001; ns, not significant compared to the control. Abbreviations: MIC, minimum inhibitory concentration; DiBAC4(3), bis(1,3-dibutylbarbituric acid) trimethine oxonol.

**Figure 3 jof-08-00901-f003:**
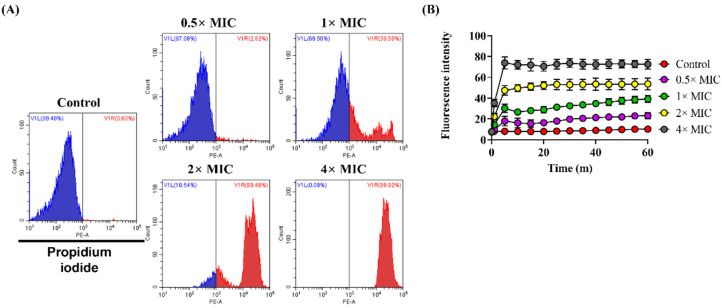
Analysis of *Candida albicans* membrane integrity after incubation with Lycosin-II. Lycosin-II at 0.5×, 1×, 2×, and 4× MICs were incubated with *C. albicans*. (**A**) A flow cytometry analysis of membrane permeabilization using PI (10 μg/mL) in *C. albicans*. (**B**) The kinetics of the membrane disruption of *C. albicans* were confirmed using SYTOX green (1 μM), as well as by monitoring the fluorescence intensity. Abbreviations: PI, propidium iodide; MIC, minimum inhibitory concentration.

**Figure 4 jof-08-00901-f004:**
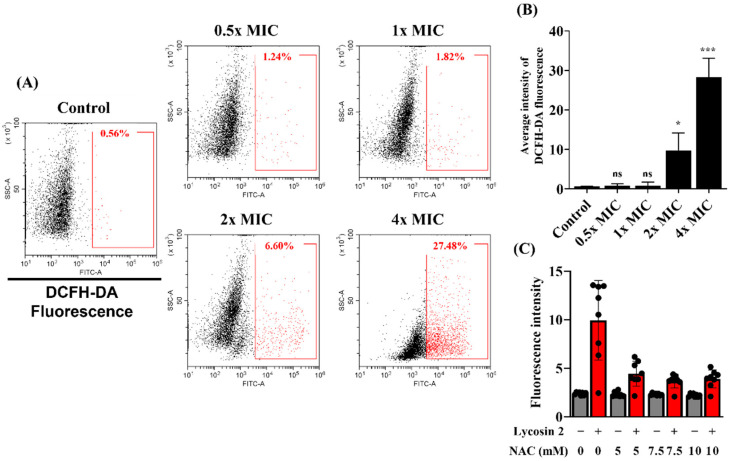
Analysis of reactive oxygen species (ROS) induced by Lycosin-II in *Candida albicans*. (**A**) ROS production in *C. albicans* added to Lycosin-II was measured using DCFH-DA detection and flow cytometry. (**B**) The statistical analyses are displayed as averages (*n* = 3). (**C**) The effect of N-acetyl cysteine (NAC) at different concentrations on membrane integrity using SYTOX green. Black dot indicated fluorescence intensity of SYTOX green. The graph is expressed as the mean ± standard deviation. Significance: *, *p* < 0.05; ***, *p* < 0.001; ns, not significant compared to the control. Abbreviations: DCFH-DA, 2, 7-dichloroflurorescin diacetate; MIC, minimum inhibitory concentration.

**Figure 5 jof-08-00901-f005:**
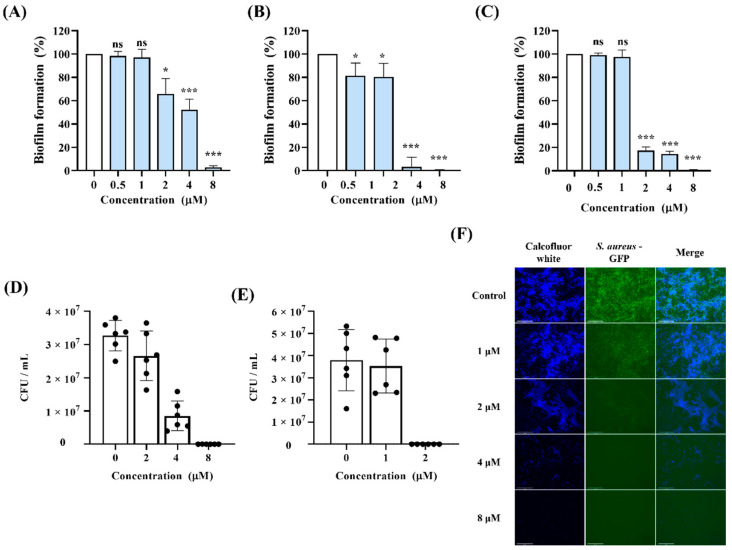
The inhibitory effect of Lycosin-II on biofilms formed by one or two microbial species. Biofilms formed by a single strain, (**A**) *Candida albicans* (CCARM 14020) or (**B**) *Staphylococcus aureus* (ATCC 25922), and (**C**) dual-species biofilms containing *C. albicans* and *S. aureus* were incubated with Lycosin-II. The biofilm mass was quantified by using crystal violet and measuring the optical density at 595 nm. The colony-forming unit (CFU) numbers of (**D**) *C. albicans* and (**E**) *S. aureus* in the dual-species biofilms were analyzed using a plating method on yeast extract peptone dextrose and Luria–Bertani medium supplemented with vancomycin and amphotericin B, respectively. Black dot indicated number of *C. albicans* and *S. aureus*. (**F**) The viability of the dual-species biofilm was analyzed using fluorescence microscopy. *C. albicans* was stained using calcofluor white (blue), and *S. aureus* expressing green fluorescent protein (GFP; green) was used to detect the bacterial cells. The graph is expressed as the mean *±* standard deviation. Significance: *, *p* < 0.05; ***, *p* < 0.001; ns, not significant compared with control.

**Table 1 jof-08-00901-t001:** Minimum inhibitory concentrations (MICs) of Lycosin-II and antifungal agents against *Candida albicans*.

Strains	MIC (μM)			
	Lycosin-II	Melittin	Fluconazole	Amphotericin B
*C. albicans* (KCTC * 7270)	2	2	16	1
*C. albicans* (CCARM * 14001)	2	2	>32	0.5
*C. albicans* (CCARM 14004)	2	4	>32	0.5
*C. albicans* (CCARM 14020)	2	2	>32	1

* Abbreviations: KCTC, Korea Collection for Type Culture; CCARM, Culture Collection of Antibiotics Resistant Microbes.

## Data Availability

The data showed in this study are available on request from the corresponding author.

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
