# Peer review of "Lycosin-II Exhibits Antifungal Activity and Inhibits Dual-Species Biofilm by Candida albicans and Staphylococcus aureus"

_jof, 2022, doi:10.3390/jof8090901_

Round 1
Reviewer 1 Report
The authors study the role of the antimicrobial peptide lycosin-II against Candida albicans and against the biofilm of Candida albicans and Staphylococcus aureus. It is an interesting work and supported with several experiments, but the authors have indicated "we investgated antifungal activity of Lycosin- II against C. albicans as well as cytotoxicity against HaCaT cells derived from Human kerationcytes" and the study of the citotoxicity against the cells has not been done. Also, an extensive editing of English language is required to be accepted for publication. There are many mistakes such as:
-line 66: investigated
-line 68: the name of fungi and bacteria with italics
-line 74: Sigma
-line 75: microorganisms
-line 91: after a comma, it is lowercase
-line 126: measurement
-line 133: analysis
And there are a lot of mistakes like those.
Author Response
Reviewer 1
- The authors study the role of the antimicrobial peptide Lycosin-II against Candida albicansand against the biofilm of Candida albicansand Staphylococcus aureus. It is an interesting work and supported with several experiments, but the authors have indicated "we investigated antifungal activity of Lycosin- II against C. albicans as well as cytotoxicity against HaCaT cells derived from Human keratinocytes" and the study of the cytotoxicity against the cells has not been done.
Response: We thank the reviewer for pointing this out. We are sorry for the confusion and we have removed the sentence “the cytotoxicity against HaCaT cells derived from human keratinocytes”. Our groups have already evaluated the cytotoxicity of Lyosin-II against Hs27 cells isolated from human foreskin fibroblasts and its hemolytic activity against sheep red blood cells [1].
(Reference)
[1] Oh, J.H.; Park, J.; Park, Y. Anti-biofilm and anti-inflammatory effects of lycosin-ii isolated from spiders against multi-drug resistant bacteria. Biochimica et Biophysica Acta (BBA)-Biomembranes 2022, 1864, 183769.
- Also, an extensive editing of English language is required to be accepted for publication. There are many mistakes such as: -line 66: investigated, line 68: the name of fungi and bacteria with italics line 74: Sigma, line 75: microorganisms, line 91: after a comma, it is lowercase, line 126: measurement, line 133: analysis, and there are a lot of mistakes like those.
Response: We thank the reviewer for pointing this out. We apologize for these errors. We have entirely corrected the sentences and typographical errors.
Reviewer 2 Report
The article presents interesting data on the inhibition of mixed biofilm formation of C. albicans and S.aureus by AMPs. The research group has several publications on the field, sometimes by using a microorganism model, sometimes by using another one. However, the manuscript could be better adjusted to make easer the reading and data presentation, being clearer and more objective. The manuscript must be carefully revised, including an extensively English revision, to avoid and correct English mistakes, such “multispeices” (line 63), “Stapylococcus aureus” (line 45), “Anlaysis” (line 133), “sigle” (line 141) as well as correction of italicized words when necessary (e.g., “C. albicans”, lines 68, 114). Abbreviation of genus and species (e.g. C. albicans – line 34) is not required.
Lines 35-36: to refer mortality rates, please use appropriate reference.
Additional comments:
Material and methods:
- Please rewrite this section including additional details of the material used/acquired (e.g. “2.1 material”, and 2.3 “Peptide synthesis and purification and Mass analysis”).
- Describe first the isolates (2.1), and then and methods (2.2, ….). Additionally, include the characteristics of the reference strains, e.g.: drug-resistant C. albicans (CCARM 14001, CCARM 14007, and CCARM 140020). The use and reason for selecting part of the isolates for the different tests performed must be clear.
- 2.3 Antifungal activity: “broth dilution method” or broth microdilution method? In addition, the authors used Yang Y, et al. Frontiers in microbiology 2020, as a reference, but checking this publication does not see the original method, once Yang Y and col. refers Yang Z (J. Med. Chem 2019), as the reference method. Please check and correct. The strains had their MICs measured by using the additional standard method as CLSI or EUCAST? Why were the MICs determined by using 600 nm once both articles already mentioned used 492 nm? How the inhibitory concentration for antifungal, peptide, and melittin were determined?
- 2.6 include the meaning of PI
Results
The authors make a brief but elucidating explanation of the importance of the method used to answer the questions of the work. I suggest placing such explanations in each of the methods, not in the results.
Discussion
The impact of the results is not fully explored. What is the association between membrane-depolarizing activity against C. albicans by Lycosin-II?
References
Should be adjusted to avoid using the previous publications from the same group including the original ones.
Author Response
Reviewer 2
- The article presents interesting data on the inhibition of mixed biofilm formation of C. albicansand S. aureusby AMPs. The research group has several publications on the field, sometimes by using a microorganism model, sometimes by using another one. However, the manuscript could be better adjusted to make easer the reading and data presentation, being clearer and more objective.
Response: We thank the reviewer for pointing this out. We apologize for these errors. We have edited the sentences to make for easier reading.
- The manuscript must be carefully revised, including an extensively English revision, to avoid and correct English mistakes, such “multispeices” (line 63), “Stapylococcus aureus” (line 45), “Anlaysis” (line 133), “sigle” (line 141) as well as correction of italicized words when necessary (e.g., “C. albicans”, lines 68, 114).
Response: We thank the reviewer for pointing this out. We have corrected typos and changed italicized words for C. albicans and S. aureus in the manuscript.
- Abbreviation of genus and species (e.g. C. albicans– line 34) is not required.
Response: We thank the reviewer for pointing this out. We have referred to the full species names for C. albicans and S. aureus in the Abstract (lines 16 and 21) and removed abbreviations in lines 34 and 46.
(after revision).
Line 16 C. albicans → Candida albicans
Line 35 Candida albicans (C. albicans) → Candida albicans
Line 21 S. aureus → Staphylococcus aureus
Line 46 Staphylococcus aureus (S. aureus) → Staphylococcus aureus
- Lines 35-36: to refer mortality rates, please use appropriate reference.
Reply: We thank the reviewer for raising this point to improve the clarity of the Introduction. We agree with the reviewer’s comments. According to a report by Lee et al. (2020), C. albicans is classified as a nosocomial pathogen that causes bloodstream infections in the United States and contributes to 40% mortality rates. Chen et al. (2020) also mentioned that C. albicans has served as the leading causal agent of life-threatening invasive infections, with mortality rates approaching 40% despite treatment. To address the reviewer’s comment, we have cited three previously reported publications.
(after revision).
(Line 35)
- albicans can cause nosocomial bloodstream infections with a mortality rate of 40%, despite the use of antifungal agents [5-7].
(References)
- Lee, Y.; Puumala, E.; Robbins, N.; Cowen, L.E. Antifungal drug resistance: Molecular mechanisms in candida albicans and beyond. Chemical reviews 2020, 121, 3390-3411.
- Chen, H.; Zhou, X.; Ren, B.; Cheng, L. The regulation of hyphae growth in candida albicans. Virulence 2020, 11, 337-348.
- Nile, C.; Falleni, M.; Cirasola, D.; Alghamdi, A.; Anderson, O.F.; Delaney, C.; Ramage, G.; Ottaviano, E.; Tosi, D.; Bulfamante, G. Repurposing pilocarpine hydrochloride for treatment of candida albicans infections. Msphere 2019, 4, e00689-00618.
- Please rewrite this section including additional details of the material used/acquired (e.g. “2.1 material”, and 2.3 “Peptide synthesis and purification and Mass analysis”).
Reply: We thank the reviewer for raising this point and making suggestions for improving the clarity of the methods. We have provided detailed information in the Materials and Methods.
(after revision).
(Line 70)
2.1 Materials
Yeast extract dextrose peptone (YPD) broth and Luria–Bertani (LB) broth were purchased from LPS Solution (Daejeon, South Korea). Agar and crystal violet was purchased from Duksan (Ansan, Korea). Rosewell Park Memorial Institute (RPMI) 1640 medium was purchased at Welgene (Daegu, Korea). Calcofluor white, bis(1,3-dibutylbarbituric acid) trimethine oxonol (DiBAC4(3)), 2, 7- Dichlorofluorescin diacetate (DCFH-DA), SYTOX Green, N-acetyl cysteine (NAC), Propidium iodide (PI) fluconazole, and amphotericin B were obtained from Sigma-Aldrich (St. Louis, MO, USA).
- albicans (KCTC 7270), used as a reference strain, was isolated from human skin lesion of erosion interdigitalis in Uruguay and was purchased from the KCTC (Korea Collection for Type Culture). C. albicans (CCARM 14001), C. albicans (CCARM 14004), C. albicans (CCARM 14007), isolated in 1999, and C. albicans (CCARM 14020), isolated in 2002, were used as representative drug-resistant strains and were purchased from the CCARM (Culture Collection of Antibiotics Resistant Microbes) [17]. S. aureus (ATCC 25923) was purchased from the American Type Culture Collection (ATCC).
2.2 Peptide synthesis and purification and Mass analysis
Lycosin-II and melittin were synthesized, purified, and analyzed as previously described [16]. Briefly, peptides were synthesized using solid-phase methods with Fmoc (N-(9-fluorenyl)methoxycarbonyl)-protected amino acids in a Liberty microwave peptide synthesizer (CEM, Co., Matthews, NC, USA). Coupling was initiated with 0.1 M N-hydroxy benzotriazole (in piperidine), dimethylformamide, 0.45 M 2-(1H-benzotriazole-1-yil)-1,1,3,3-tetramethyluronium hexafluorophosphate (in dimethylformamide) and 2 M N,N-diisopropylethylamine (in N-methyl-pyrrolidone). The peptides were purified by reversed-phase high-performance liquid chromatography (RP-HPLC) on Jupiter C18 column (250 × 21.2 Zmm, 15 μM, 300 Å; Phenomenex, Torrance, CA, USA). The molecular weight of the peptides was analyzed using a matrix-assisted laser desorption ionization (MALDI) mass spectrometer (Kratos Anlytical Ins.).
- Describe first the isolates (2.1), and then and methods (2.2, ….). Additionally, include the characteristics of the reference strains, e.g.: drug-resistant C. albicans(CCARM 14001, CCARM 14007, and CCARM 140020). The use and reason for selecting part of the isolates for the different tests performed must be clear.
Reply: We thank the reviewer for raising this point and improving the clarity of the materials and methods. We have added the characteristics of C. albicans to the manuscript and the references. C. albicans (KCTC 7270) was used as the reference strain. Therefore, we used C. albicans (KCTC 7270) to investigate the antifungal mechanism of lycosin-II against C. albicans. Moreover, we used C. albicans (CCAR 14020) for the anti-biofilm activity of Lycosin-II because we confirmed that C. albicans (CCARM 14020) can effectively form biofilms, as mentioned in line 167.
(after revision).
(Line 77)
- albicans (KCTC 7270), used as a reference strain, was isolated from human skin lesion of erosion interdigitalis in Uruguay and was purchased from the KCTC (Korea Collection for Type Culture). C. albicans (CCARM 14001), C. albicans (CCARM 14004), C. albicans (CCARM 14007), isolated in 1999, and C. albicans (CCARM 14020), isolated in 2002, were used as representative drug-resistant strains and were purchased from the CCARM (Culture Collection of Antibiotics Resistant Microbes) [17]. S. aureus (ATCC 25923) was purchased from the American Type Culture Collection (ATCC).
References
- Park, S.-C.; Kim, J.-Y.; Kim, E.-J.; Cheong, G.-W.; Lee, Y.; Choi, W.; Lee, J.R.; Jang, M.-K. Hydrophilic linear peptide with histidine and lysine residues as a key factor affecting antifungal activity. International Journal of Molecular Sciences 2018, 19, 3781.
- 2.3 Antifungal activity: “broth dilution method” or broth microdilution method? In addition, the authors used Yang Y, et al. Frontiers in microbiology 2020, as a reference, but checking this publication does not see the original method, once Yang Y and col. refers Yang Z (J. Med. Chem 2019), as the reference method. Please check and correct. The strains had their MICs measured by using the additional standard method as CLSI or EUCAST? Why were the MICs determined by using 600 nm once both articles already mentioned used 492 nm? How the inhibitory concentration for antifungal, peptide, and melittin were determined?
Reply: We thank the reviewer for raising this point for improving the clarity of the materials, methods, and questions. First, we revised the references appropriately for the method used in the manuscript. We did not use these two methods (CLSI or EUCAST). Rather, we followed the standard broth microdilution method and have included a reference for this method. The latter method is widely used to compare the activities of different antimicrobial peptides. The microdilution test can be useful for screening the antifungal activity of a target peptide using a relatively small amount of the peptide. At the same time, peptides can be tested at a range of concentrations. Therefore, this experiment is an important and widely used first step in screening for a new antifungal peptide.
Optical density at 600 nm was used to investigate the growth of C. albicans. The reason for using an OD of 600 nm was to minimize cell damage and growth. In many published papers, the OD at 600 nm is used to determine the MICs values. We determined that the lowest peptide and antifungal agent concentrations inhibited 90% of the growth of C. albicans compared to the negative control.
(after revision).
(Line 98)
To determine the antifungal activity of Lycosin-II, melittin, and antifungal agents, the minimum inhibitory concentrations (MICs) were determined using the broth microdilution method with some modification [18,19]. Lycosin-II, melittin, fluconazole, and amphotericin B were serially diluted two-fold in 10 mM sodium phosphate buffer in 96 well plates. C. albicans was cultured overnight at 28 °C in YPD media, and then seeded into 96-well plates (2 × 104 colony forming units (CFUs)/mL - 50 μL/well). After incubation for 16-24 h at 28 °C, the growth of the C. albicans strains was determined by measuring the absorbance at 600 nm using a Spectra Max M3 microplate reader. The MICs were determined as the lowest peptide or antifungal agent concentration that can inhibit 90% of the growth of C. albicans compared to negative control.
(Reference)
- Radhakrishnan, V.S.; Mudiam, M.K.R.; Kumar, M.; Dwivedi, S.P.; Singh, S.P.; Prasad, T. Silver nanoparticles induced alterations in multiple cellular targets, which are critical for drug susceptibilities and pathogenicity in fungal pathogen (candida albicans). International journal of nanomedicine 2018, 13, 2647.
- Hong, M.J.; Kim, M.K.; Park, Y. Comparative antimicrobial activity of hp404 peptide and its analogs against acinetobacter baumannii. International Journal of Molecular Sciences 2021, 22, 5540.
- 2.7 include the meaning of PI
Reply: Thank you for this suggestion. We have added detailed information on PI.
(after revision).
(Line 128)
The membrane integrity of C. albicans (KCTC 7270) after exposure to Lycosin-II was determined using a PI uptake assay. PI is impermeable dye used to determine permeability of cells by agents [22,23].
- Results - The authors make a brief but elucidating explanation of the importance of the method used to answer the questions of the work. I suggest placing such explanations in each of the methods, not in the results.
Reply: We thank the reviewer for raising this point and improving this manuscript. We have transferred the description of fluorescence dyes used in the manuscript from the results to the methods section.
(after revision).
(Line 118)
2.5 Membrane depolarization assay
Membrane depolarization in C. albicans (KCTC 7270) by Lycosin-II was measured using DiBAC4(3) [21]. DiBAC4(3) is a membrane potential-sensitive dye that can enter depolarized cells and increase fluorescence intensity. Briefly, C. albicans was cultured in YPD medium at 28 °C and then washed with phosphate buffered saline (PBS) three times. C. albicans (5â…¹106 CFU/mL) in PBS was incubated with Lycosin-II at 0.5â…¹, 1â…¹, 2â…¹ and 4â…¹MIC for 3 h at 28 °C. The cells were centrifuged and resuspended in 1 mL of PBS supplemented with 2 μg/mL DiBAC4(3) for 10 min. The fluorescence intensity of DiBAC4(3) was measured by flow cytometry (Beckman, Brea, CA, USA).
(Line 127)
2.6 PI uptake assay
The membrane integrity of C. albicans (KCTC 7270) after exposure to Lycosin-II was determined using a PI uptake assay. PI is impermeable dye used to determine permeability of cells by agents [22,23].
(Line 137)
2.7 SYTOX green uptake assay
To further investigate C. albicans membrane damage by Lycosin-II, we used SYTOX green which is an impermeable dye that interacts with nucleic acids resulting in enhanced fluorescent intensity. .
(Line 146)
2.8 Measurement of ROS
The accumulation ROS in C. albicans (KCTC 7270) by due to Lycosin-II was detected using DCFH-DA [25]. DCFH-DA is oxidized by ROS to dichlorofluorescein (DCF), which emits green fluorescence [26].
- Discussion - The impact of the results is not fully explored. What is the association between membrane-depolarizing activity against C. albicans by Lycosin-II?
(after revision).
Response: We thank the reviewer for raising this question. DiBAC4(3), which is a membrane potential-sensitive dye, enters depolarized cells and interacts with intercellular proteins or membranes, resulting in an increase in fluorescence intensity. Therefore, we used DiBAC4(3) to investigate membrane depolarization in C. albicans following exposure to lycosin-II. In C. albicans exposed to lycosin-II at 0.5x, 1x, 2x and 4x MIC, DiBAC4(3) fluorescence intensity increased by 5.06%, 24.82%, 90.44%, and 99.04%, respectively. However, the fluorescence intensity of DiBAC4(3) barely increased in untreated C. albicans. These results indicate that lycosin-II induces changes in the membrane potential of C. albicans.
- References- Should be adjusted to avoid using the previous publications from the same group including the original ones.
Reply: Thank you for this suggestion. We agree with the reviewer’s comments. We have added references published by other groups.
(after revision).
(Line 127)
2.6 PI uptake assay
The membrane integrity of C. albicans (KCTC 7270) after exposure to Lycosin-II was determined using a PI uptake assay. PI is impermeable dye used to determine permeability of cells by agents [22,23].
References
- Seyedjavadi, S.S.; Khani, S.; Eslamifar, A.; Ajdary, S.; Goudarzi, M.; Halabian, R.; Akbari, R.; Zare-Zardini, H.; Imani Fooladi, A.A.; Amani, J. The antifungal peptide mch-amp1 derived from matricaria chamomilla inhibits candida albicans growth via inducing ros generation and altering fungal cell membrane permeability. Frontiers in microbiology 2020, 10, 3150.
(Line 163)
2.10 Biofilm inhibition assays
To analyze the effect of Lycosin-II on single-species biofilms of C. albicans and polymicrobial biofilms, we used C. albicans (CCARM 140020), S. aureus (ATCC 25923), and S. aureus (ATCC 25923) expressing green fluorescent protein (GFP). In a previous study, we demonstrated that the C. albicans strain CCARM 140020 can form a large biofilm [28,29].
References
- Luo, Y.; McAuley, D.F.; Fulton, C.R.; Sá Pessoa, J.; McMullan, R.; Lundy, F.T. Targeting candida albicans in dual-species biofilms with antifungal treatment reduces staphylococcus aureus and mrsa in vitro. PloS one 2021, 16, e0249547.
(Line 176)
Ethanol (95%) was then added to the stained biofilms for dissolution. Biofilm quantification was determined by measuring absorbance at 595 nm using a Versa-Max microplate enzyme-linked immunosorbent assay (ELISA) reader (Molecular Devices, Sunnyvale, CA, USA) [30,31].
References
- Wang, Y.; Pei, Z.; Lou, Z.; Wang, H. Evaluation of anti-biofilm capability of cordycepin against candida albicans. Infection and Drug Resistance 2021, 14, 435.
Reviewer 3 Report
In this study, authors investigate the antifungal and antibiofilm activity of the antimicrobial peptide lycosin-II, evaluating the possible mechanism of action (membrane damage and ROS production). However, the text is poorly written, with several English mistakes.
Revise English.
Abstract should be better structured. Methods describe only the effect against Candida albicans, while Results report effect against C. albicans with Staphylococcus aureus.
Abbreviations should be described in full at their first appearance.
Introduction
Revise English, several sentences show grammar problems.
Methods
Line 114: who long was the incubation?
Lines 120-125: what is the purpose of the SYTOX green uptake assay? Is it membrane permeability similar to propidium iodide?
Lines 138-139: was fluorescence measured using a spectrophotometer? How it is possible?
Line 158: how was the biofilm removed for viability assay?
Why authors did not evaluate pre-formed biofilms?
Methods section lacks information of data and statistical analysis.
Results
There are several errors of grammar, spelling, sentences, etc.
Discussion
Line 301: “MDR” in full.
Results are not discussed with others described in the literature. Several parts of discussion seem to be Introduction (explaining the problem addressed by the research). The focus should be on the findings of the investigation.
Author Response
Reviewer3
- In this study, authors investigate the antifungal and antibiofilm activity of the antimicrobial peptide lycosin-II, evaluating the possible mechanism of action (membrane damage and ROS production). However, the text is poorly written, with several English mistakes.
Revise English. Abstract should be better structured.
Response: We thank the reviewer for pointing this out. We apologize for the confusion. We have corrected the sentences and typographical errors.
- Methods describe only the effect against Candida albicans, while Results report effect against C. albicans with Staphylococcus aureus.
Response: We thank the reviewer for pointing this out. We only used S. aureus in the biofilm inhibition assays. Briefly, we confirmed that lycosin-II effectively inhibits dual-species biofilm formation by C. albicans and S. aureus using crystal violet staining and fluorescence microscopy. We marked in red in relation to S. aureus.
2.10 Biofilm inhibition assays
To analyze the effect of Lycosin-II on single-species biofilms of C. albicans and polymicrobial biofilms, we used C. albicans (CCARM 140020), S. aureus (ATCC 25923), and S. aureus (ATCC 25923) expressing green fluorescent protein (GFP). In a previous study, we demonstrated that the C. albicans strain CCARM 140020 can form a large biofilm [28,29]. C. albicans (CCARM 140020) and S. aureus (ATCC 25923) were incubated in YPD and LB media. The two strains were washed with RPMI medium supplemented with 2% glucose and adjusted to 1â…¹106 CFU/mL for single-species biofilm evaluation. Aliquots (180 μL) of C. albicans or S. aureus and the two mixed strains were added into 96-well plates and incubated with 20 μL of Lycosin-II (0.5 μM-8 μM for 24 h at 37 °C. The supernatant was discarded and 100% methanol was added to fix the biofilm for 20 min. Methanol was discarded, and the sample was completely dried. The biofilm formation was stained with 0.1% crystal violet for 20 min and then washed three times with distilled water. Ethanol (95%) was then added to the stained biofilms for dissolution. Biofilm quantification was determined by measuring absorbance at 595 nm using a Versa-Max microplate enzyme-linked immunosorbent assay (ELISA) reader (Molecular Devices, Sunnyvale, CA, USA) [30,31]. The percentage biofilm formation was determined using the following equation:
Biofilm formation (%) = (A595 of treated biofilm/A595 of untreated biofilm) × 100
2.11 Determination of C. albicans and S. aureus viability in dual species biofilm
The biofilms formed by C. albicans and S. aureus were suspended by pipetting. The biofilm suspension was transferred to a 1.5 mL tube, diluted in 1 mL PBS, and vortexed for 5 min [32, 33]. YPD and LB agar plates containing vancomycin and amphotericin B, respectively, were used to distinguish between C. albicans and S. aureus. Aliquots were spread onto agar plates and incubated at 37 °C for 16 h. The numbers of C. albicans and S. aureus were determined by counting CFUs.
2.12 Visualization of biofilm
To visualize the biofilm after treatment with ycosin-II, we used calcofluor white (10 μg/mL) to stain C. albicans in the biofilm. Moreover, the S. aureus (ATCC 25923) strain expressing GFP allowed the visualization of this bacterium. Images were visualized using EVOS FL Auto 2 fluorescence (Invitrogen, Waltham, MA, USA).
- Abbreviations should be described in full at their first appearance.
Response: We thank the reviewer for pointing this out. We have added abbreviations in full at the first appearance and used abbreviations.
First abbreviations
|
Line 11 |
Antimicrobial peptides (AMPs) |
|
Line 16 |
Candida albicans (C. albicans) |
|
Line 19 |
Reactive oxygen species (ROS) |
|
Line 21 |
Staphylococcus aureus (S. aureus) |
|
Line 41 |
Extracellular polymeric substances (EPS) |
|
Line 70 |
Yeast extract dextrose peptone (YPD) |
|
Line 70 |
Luria-Bertani (LB) |
|
Line 72 |
Rosewell Park Memorial Institute (RPMI) |
|
Line 73 |
Bis(1,3-dibutylbarbituric acid) trimethine oxonol ((DiBAC4(3)) |
|
Line 74 |
2, 7-Dichlorofluorescin diacetate (DCFH-DA) |
|
Line 75 |
N-acetylcysteine (NAC) |
|
Line 75 |
Propidium iodide (PI) |
|
Line 78 |
Korea Collection for Type Culture (KCTC) |
|
Line 81 |
Culture Collection of Antibiotics Resistant Microbes (CCARM) |
|
Line 98 |
Minimum inhibitory concentrations (MICs) |
|
Line 102 |
Colony forming units (CFUs) |
|
Line 148 |
Dichlorofluorescein (DCF) |
|
Line 166 |
Green fluorescent protein (GFP) |
- Introduction - Revise English, several sentences show grammar problems.
Response: We thank the reviewer for pointing this out. We apologize for the inconvenience in reading the manuscript. We have corrected the sentences accordingly.
- Methods- Line 114: who long was the incubation?
Reply: We thank the reviewer for raising this point and improving the clarity of the methods. We have added the incubation time.
(after revision).
(Line 131)
- albicans (2â…¹107 CFU/mL) in PBS was incubated with lycosin-II at 0.5â…¹, 1â…¹, 2â…¹ and 4â…¹MIC at 28 °C for 10 min.
- Lines 120-125: what is the purpose of the SYTOX green uptake assay? Is it membrane permeability similar to propidium iodide?
Reply: We thank the reviewer for this comment. Propidium iodide and SYTOX green did not enter the intact membrane. However, when the membrane is damaged by antimicrobial agents, they enter the cell and bind to nucleic acids, resulting in an increase in the fluorescence intensity. We have added the purpose of the SYTOX Green Uptake Assay.
(after revision).
(Line 129)
PI is impermeable dye and used to determine permeability of cells by agents.
(Line 138)
To further investigate C. albicans membrane damage by lycosin-II, we used SYTOX green, which is an impermeable dye that interacts with nucleic acids, resulting in enhanced fluorescence intensity.
- Lines 138-139: was fluorescence measured using a spectrophotometer? How it is possible?
Reply: We thank the reviewer for this comment. A Spectramax M3 spectrophotometer was used to detect the fluorescence intensity of SYTOX green at excitation and emission wavelengths of 485 nm and 520 nm, respectively. We have added the wavelength of detection to SYTOX green in the Methods section.
(after revision).
(Line 160)
The fluorescence intensity of SYTOX green was measured at excitation and emission wavelengths of 485 nm and 520 nm, respectively, using a Spectramax M3 spectrophotometer (Molecular Devices, Sunnyvale, CA, USA).
- Line 158: how was the biofilm removed for viability assay?
Reply: We thank the reviewer for this comment. First, planktonic cells were removed by washing with PBS. To measure cell viability in the biofilms, biofilms were detached by vigorous pipetting. The cells in the biofilm were then suspended in 1 mL and vortexed for 5 min. We have added this information and references.
(after revision).
The biofilms formed by C. albicans and S. aureus were suspended by pipetting. The biofilm suspension was transferred to a 1.5 mL tube, diluted in 1 mL PBS, and vortexed for 5 min [32,33].
(References)
- Cruz, C.D.; Shah, S.; Tammela, P. Defining conditions for biofilm inhibition and eradication assays for gram-positive clinical reference strains. BMC microbiology 2018, 18, 1-9.
- Galdiero, E.; Di Onofrio, V.; Maione, A.; Gambino, E.; Gesuele, R.; Menale, B.; Ciaravolo, M.; Carraturo, F.; Guida, M. Allium ursinum and allium oschaninii against klebsiella pneumoniae and candida albicans mono-and polymicrobic biofilms in in vitro static and dynamic models. Microorganisms 2020, 8, 336.
- Why authors did not evaluate pre-formed biofilms?
Reply: We have already evaluated the effect of lycosin-II on biofilm formation by C. albicans and S. aureus. This showed that Lycosin-II at high concentrations was able to eradicate biofilms. Therefore, we wanted to test lower concentrations and make analog peptides of lycosin-II in future studies. Please refer to the following figure.
10 Methods section lacks information of data and statistical analysis.
Response: We thank the reviewer for pointing this out. We have added information regarding the methods and statistical analysis.
(after revision).
(Line 84)
2.2 Peptide synthesis and purification and Mass analysis
Lycosin-II and melittin were synthesized, purified, and analyzed as previously described [16]. Briefly, peptides were synthesized using solid-phase methods with Fmoc (N-(9-fluorenyl)methoxycarbonyl)-protected amino acids in a Liberty microwave peptide synthesizer (CEM, Co., Matthews, NC, USA). Coupling was initiated with 0.1 M N-hydroxy benzotriazole (in piperidine), dimethylformamide, 0.45 M 2-(1H-benzotriazole-1-yil)-1,1,3,3-tetramethyluronium hexafluorophosphate (in dimethylformamide) and 2 M N,N-diisopropylethylamine (in N-methyl-pyrrolidone). The peptides were purified by reversed-phase high-performance liquid chromatography (RP-HPLC) on Jupiter C18 column (250 × 21.2 Zmm, 15 μM, 300 Å; Phenomenex, Torrance, CA, USA). The molecular weight of the peptides was analyzed using a matrix-assisted laser desorption ionization (MALDI) mass spectrometer (Kratos Anlytical Ins.).
(Line 118)
2.5 Membrane depolarization assay
Membrane depolarization in C. albicans (KCTC 7270) by Lycosin-II was measured using DiBAC4(3) [21]. DiBAC4(3) is a membrane potential-sensitive dye that can enter depolarized cells and increase fluorescence intensity.
(Line 127)
2.6 PI uptake assay
The membrane integrity of C. albicans (KCTC 7270) after exposure to Lycosin-II was deter-mined using a PI uptake assay. PI is impermeable dye used to determine permeability of cells by agents [22,23].
(Line 137)
2.7 SYTOX green uptake assay
To further investigate C. albicans membrane damage by Lycosin-II, we used SYTOX green which is an impermeable dye that interacts with nucleic acids resulting in enhanced fluorescent intensity.
(Line 146)
2.8 Measurement of ROS
The accumulation ROS in C. albicans (KCTC 7270) by due to Lycosin-II was detected using DCFH-DA [25]. DCFH-DA is oxidized by ROS to dichlorofluorescein (DCF), which emits green fluorescence [26].
(Line 194)
2.13 Statistics
All data were expressed as three independent experiments using the mean ± standard deviation (SD). *, p < 0.05; **, p < 0.01; ***, p < 0.001; ****, p < 0.0001; ns, not significant compared with control.
- Results-There are several errors of grammar, spelling, sentences, etc.
Response: We thank the reviewer for pointing this out. We apologize for these errors. We have corrected the sentences and typographical errors.
- Discussion Line 301: “MDR” in full.
Reply: We thank the reviewer for raising this point. We added MDR
(after revision).
(Line 330)
Multidrug-resistant (MDR)-S. aureus and MDR-P. aeruginosa
- The results have not been discussed with those described in the literature. Several parts of the discussion seem to be in the Introduction (explaining the problem addressed by the research). The focus of this study should be on the findings.
Response: We thank the reviewer for pointing this out. As per the reviewer’s suggestion, we have added a discussion about previous studies, as described in the literature.
(after revision).
(Line 345)
In a previous study, Bac8c, the peptide analogue of Ba2A, induced a change in the membrane potential in C. albicans [43].
(Line 352)
Previous studies reported that the membrane of C. albicans was damaged by antifungal peptides such as protonectin isolated from Agelaia pallipes pallipes [34]. In contrast, peptides consisting of lysine and tryptophan 4 repeats showed antifungal activity against C. albicans but did not affect membrane integrity [23]. To explore the antifungal mechanism of Lycosin-II on the C. albicans membrane,…

Round 2
Reviewer 1 Report
After corrections made by the authors, the article has improved and is suitable for publication in the Journal of Fungi.
Author Response
After corrections made by the authors, the article has improved and is suitable for publication in the Journal of Fungi.
Reply: Thank you for your previous comments that helped us improve this manuscript.
Reviewer 2 Report
Authors answered the questions performed by the reviewers, but the statistical methods used during the analysis is not elucidated.
Author Response
Authors answered the questions performed by the reviewers, but the statistical methods used during the analysis is not elucidated.
Reply: We thank the reviewer for raising this point and improving this manuscript. We added explanation of statistics.
(after revision)
Line 195
2.13 Statistics
All data were expressed as three independent experiments using the mean ± standard deviation (SD). Two-tailed student’s t-tests were used to determine the significance of the differences. *, p < 0.05; **, p < 0.01; ***, p < 0.001; ****, p < 0.0001; ns, not significant compared with control.
Reviewer 3 Report
Authors answered all questions made by the reviewers.
However, in the statistical analysis, authors provided only, the p values considered significant. They do not mention which statistical tests were used for analysis.
Minor English revision is still needed.
Author Response
Authors answered all questions made by the reviewers. However, in the statistical analysis, authors provided only, the p values considered significant. They do not mention which statistical tests were used for analysis. Minor English revision is still needed.
Reply: We thank the reviewer for raising this point and improving this manuscript. We added explanation of statistics. Moreover, we entirely corrected the sentences.
(after revision)
Line 195
2.13 Statistics
All data were expressed as three independent experiments using the mean ± standard deviation (SD). Two-tailed student’s t-tests were used to determine the significance of the differences. *, p < 0.05; **, p < 0.01; ***, p < 0.001; ****, p < 0.0001; ns, not significant compared with control.